# Ten-Week Sucralose Consumption Induces Gut Dysbiosis and Altered Glucose and Insulin Levels in Healthy Young Adults

**DOI:** 10.3390/microorganisms10020434

**Published:** 2022-02-14

**Authors:** Lucía A. Méndez-García, Nallely Bueno-Hernández, Miguel A. Cid-Soto, Karen L. De León, Viridiana M. Mendoza-Martínez, Aranza J. Espinosa-Flores, Miguel Carrero-Aguirre, Marcela Esquivel-Velázquez, Mireya León-Hernández, Rebeca Viurcos-Sanabria, Alejandra Ruíz-Barranco, Julián M. Cota-Arce, Angélica Álvarez-Lee, Marco A. De León-Nava, Guillermo Meléndez, Galileo Escobedo

**Affiliations:** 1Laboratory of Immunometabolism, Research Division, General Hospital of Mexico, Mexico City 06720, Mexico; angelica.mendez.86@hotmail.com (L.A.M.-G.); viurcos.reb@hotmail.com (R.V.-S.); 2Laboratory for Proteomics and Metabolomics, General Hospital of Mexico, Mexico City 06720, Mexico; nallely_bh5@yahoo.com.mx (N.B.-H.); karendeleonb@yahoo.com.mx (K.L.D.L.); viridiana_2909@hotmail.com (V.M.M.-M.); jhosadara.espinosa@gmail.com (A.J.E.-F.); carreroam.90@hotmail.com (M.C.-A.); esquivel.marcela@gmail.com (M.E.-V.); leonhmireya@gmail.com (M.L.-H.); 3Immunogenomics and Metabolic Diseases Laboratory, Instituto Nacional de Medicina Genómica, Mexico City 14610, Mexico; macid@inmegen.gob.mx; 4PECEM, Facultad de Medicina, Universidad Nacional Autónoma de México, Mexico City 04510, Mexico; 5Clinical Nutrition Division, General Hospital of Mexico, Mexico City 06720, Mexico; aleruba30@hotmail.com; 6Department of Biomedical Innovation, Center for Scientific Research and Higher Education of Ensenada (CICESE), Baja California 22860, Mexico; jmcotaarce@gmail.com (J.M.C.-A.); ibeth@cicese.mx (A.Á.-L.); madeleon@cicese.mx (M.A.D.L.-N.); 7Facultad de Salud Pública y Nutrición, Universidad Autónoma de Nuevo León, Monterrey 64460, Mexico

**Keywords:** sucralose, microbiome, glucose load, Firmicutes, *Blautia coccoides*, dysbiosis

## Abstract

Sucralose consumption alters microbiome and carbohydrate metabolism in mouse models. However, there are no conclusive studies in humans. Our goals were to examine the effect of sucralose consumption on the intestinal abundance of bacterial species belonging to Actinobacteria, Bacteroidetes, and Firmicutes and explore potential associations between microbiome profiles and glucose and insulin blood levels in healthy young adults. In this open-label clinical trial, volunteers randomly drank water, as a control (*n* = 20), or 48 mg sucralose (*n* = 20), every day for ten weeks. At the beginning and the end of the study, participants were subjected to an oral glucose tolerance test (OGTT) to measure serum glucose and insulin every 15 min for 3 h and provided fecal samples to assess gut microbiota using a quantitative polymerase chain reaction. Sucralose intake altered the abundance of Firmicutes without affecting Actinobacteria or Bacteroidetes. Two-way ANOVA revealed that volunteers drinking sucralose for ten weeks showed a 3-fold increase in *Blautia coccoides* and a 0.66-fold decrease in *Lactobacillus acidophilus* compared to the controls. Sucralose consumption increased serum insulin and the area under the glucose curve compared to water. Long-term sucralose ingestion induces gut dysbiosis associated with altered insulin and glucose levels during an OGTT.

## 1. Introduction

Sucralose is a non-caloric artificial sweetener (NAS) that confers a sweet taste to food and beverages without increasing calorie intake [1]. However, numerous studies have recently shown that sucralose consumption is associated with alterations in the metabolism of glucose and insulin by disrupting the balance of the gut microbiome [2,3,4].

Mice drinking sucralose for 6 months exhibit changes in gut bacteria belonging to the phylum Firmicutes [4], among which *Blautia coccoides* and *Lactobacillus acidophilus* have been shown to play key roles in insulin resistance and lipid metabolism [5,6]. Similarly, mice drinking sucralose for eleven weeks show glucose intolerance associated with an increased intestinal abundance of bacteria belonging to the phyla Bacteroidetes and Actinobacteria, such as *Bacteroides uniformis* and *Bifidobacterium longum* [2]. Conversely, the effect of sucralose consumption on gut microbiota composition still shows controversial results in humans. Adults consuming 0.136 g sucralose for two 14-day periods showed no changes in the intestinal abundance of bacteria belonging to the phyla Bacteroidetes and Actinobacteria [7]. Likewise, a clinical study conducted in male volunteers that took 780 mg sucralose for 7 days revealed no significant effects on the intestinal abundance of bacteria belonging to the phylum Firmicutes [8]. In contrast, a recent study examining fecal samples from thirteen adult volunteers reported that sucralose directly increases the abundance of intestinal bacteria belonging to the genera *Escherichia*, *Shigella*, and *Bilophila* [9]. As we have outlined here, sucralose intake appears to associate with dysbiosis-related metabolic alteration. However, the vast majority of studies have been conducted in experimental animal models and evidence in humans is still uncertain.

We conducted a clinical trial in healthy young adults to investigate the effect of long-term sucralose ingestion on the intestinal abundance of four bacterial species representative of the phyla Actinobacteria, Bacteroidetes, and Firmicutes as a primary outcome. As a secondary outcome, we explored whether sucralose-induced dysbiosis was potentially linked to alterations in glucose and insulin levels during an oral glucose load.

## 2. Materials and Methods

### 2.1. Subjects

We enrolled 47 healthy adult volunteers of both sexes, who were screened by means of the homeostasis model assessment of insulin resistance (HOMA-IR) values ≤2.5 and body mass index (BMI) 18.5–29.9 kg/m^2^, aging between 18 and 35 years in this open-label, randomized clinical trial. All study participants were not habitual consumers of sucralose-containing products and provided written informed consent to participate in the study. We conducted this trial in rigorous adherence to the principles described in the 1964 Declaration of Helsinki and its posterior amendment in 2013 (registration of the ethical code approval: DI/16/301/03/022). According to a list dispensed, participants agreed to avoid consuming any non-caloric sweeteners during the study. They also decided to follow a balanced diet consisting of vegetables, fruits, grains, proteins, and dairy that we recommended according to the physical activity level and monitored weekly via phone call interview. We also calculated the amount of protein, carbohydrates, and lipids ingested by each participant weekly. Moreover, we registered each participant’s physical activity levels weekly, which mainly consisted of running for no more than forty minutes, two or three days per week. We excluded volunteers from the study who were previously diagnosed with any of the following diseases: type 2 diabetes; hypertension; chronic liver disease; kidney disease; malabsorption disorder; short bowel syndrome; functional gastrointestinal disorders; infectious diseases. Those who had been pregnant, lactating, or underwent antibiotic therapy in the last six months were also excluded. We also considered alcohol use and smoking as exclusion criteria. Subjects who agreed to participate in the study reported they usually drank ≥1 drink per month but <1 drink per week (1 drink = 14 g pure alcohol), and they were lifelong non-smokers. We eliminated participants from the study who did not attend at least 80% of the medical consultation and weekly appointments and those who did not drink the bottle of water or sucralose at least 80% of the ten weeks the study lasted. This study contains partial results of the work registered at clinicaltrials.gov (identifier code: NCT03703141) and meets the Consolidated Standards of Reporting Trials (CONSORT) guidelines.

### 2.2. Study Design

Estimation of sample size was performed by the GPower v.3.1.9.2 program, expecting an effect size of 1.1 with an alpha error of 0.05 and a power of 95% for two independent groups, resulting in 17 individuals per group plus 20% losses for a final sample size of *n* = 20 subjects per group. The control group received bottles containing 60 mL of sterile water each. The sucralose group received bottles equivalent to 48 mg of pure liquid sucralose (American Health Foods & Ingredients, San Diego, CA, USA) dissolved in 60 mL of sterile water each. All participants had breakfast between 7 and 9 a.m. every day, and sucralose or water sips were drunk ten minutes before breakfast. Daily ingestion of 48 mg of sucralose resembles consumption of four commercial Splenda^®^ packets per day, representing less than 15% of the Acceptable Daily Intake (ADI) set by the Food and Drug Administration (FDA), and less than 5% of the ADI established by the Joint Food and Agriculture Organization (FAO)/World Health Organization Expert Committee on Food Additives (JECFA) [10,11]. At the beginning of the study, all participants were subjected to an oral glucose tolerance test (OGTT) for 180 min, using 75 g of glucose dissolved in sterile water. We obtained blood samples from all study subjects starting with oral glucose load at min zero and every 15 min for 180 min for quantifying the serum levels of glucose and insulin using an intravenous catheter closed system. After ten weeks, we performed a second OGTT in all participants to quantify the blood glucose and insulin levels again, as described before. We evaluated the adherence rate to water or sucralose consumption in the weekly appointments by counting empty bottles in a personal interview. At the study’s beginning and end, we requested all participants to bring a stool sample for bacterial DNA isolation. We stored the stool samples at −80 °C for no more than six months.

### 2.3. Bacterial Species Identification

We isolated bacterial DNA from stool samples of all study volunteers using the QI-Aamp Power Fecal DNA kit (QIAGEN, Germantown, MD, USA). DNA samples (100 ng/μL) were individually placed in the BIO-RAD CFX96 Real-Time System thermal cycler using SsoAdvanced Universal SYBR Green Supermix (Bio-Rad Laboratories, Hercules, CA, USA). A Pan-bacteria universal primer was used as a control gene for ∆∆CT calculation. Results are expressed as fold change by comparing bacterial gene abundance in triplicates at the beginning and the end of the intervention. Primers sequences used in this study are as follow: phylum Actinobacteria, species *Bifidobacterium longum*, forward 5′-CGCACATGCTCATCGAACTG-3′ and reverse 5′-GATGATGCCGTCCGAAAACG-3′ (product length 77 bp); phylum Bacteroidetes, species *Bacteroides uniformis*, forward 5′-CCAACAGCTTTTCCGGGTTG-3′ and reverse 5′-AAGGCCTTCCGAGAGGGTAT-3′ (product length 574 bp); phylum Firmicutes, species *Blautia coccoides* (formerly known as *Clostridium coccoides*), forward 5′-GTAGAGCTGATCCCGAGCAC-3′ and reverse 5′-TCGTACGATACGGAAGCAGC-3′ (product length 792 bp); phylum Firmicutes, species *Lactobacillus acidophilus*, forward 5′-GCTTTGCAAGTGTCTGGTGG-3′; reverse 5′-GATGTGCCCACGCATCAATC-3′ (product length 172 bp); Pan-bacteria universal primer, forward 5′-TCCTACGGGAGGCAGCAGT-3′ and reverse 5′-GACTACCAGGGTATCTAATCCTGTT-3′ (product length 466 bp). Quantitative polymerase chain reaction (qPCR) experiments were reported according to the Minimum Information for Publication of Quantitative Real-Time PCR Experiments (MIQE) guidelines.

### 2.4. Statistics

We evaluated the normality of data using the Shapiro–Wilk test. The primary outcome of this study was the difference in bacterial abundance. Glucose and insulin levels, before and after water or sucralose exposure, were considered the secondary outcome. We analyzed data for glucose and insulin levels from OGTT by two-tailed 2-way ANOVA with Bonferroni correction. We compared the area under the curve (AUC) of insulin and glucose before and after water or sucralose exposure by the paired Student’s T-test, or the unpaired Student’s T-test, depending on the group comparison. We compared the relative bacterial abundance by the Wilcoxon matched-pairs signed-rank test, or the Mann–Whitney U test, depending on the group comparison. We showed the results as mean ± standard deviation or box plot using the GraphPad Prism 7.0 software (GraphPad Software, La Jolla, CA 92037, USA). We used the Spearman’s correlation model to calculate coefficients (*r*) and *p* values of the changes in bacterial abundance with AUC of insulin (AUCI) and AUC of glucose (AUCG) in control and sucralose groups, using the R i386 3.5.2 terminal. We considered a difference as significant when *p* < 0.05.

## 3. Results

Figure 1 illustrates the selection process of participants enrolled in the study. After meeting inclusion and exclusion criteria, we eliminated seven participants from the study because they refused to attend at least 80% of weekly appointments. (Figure 1).

At the beginning and end of the study, participants who completed the ten-week water or sucralose consumption showed similar demographic, anthropometric, laboratory parameters, physical activity levels, and dietary nutrient intake (Table 1). However, sucralose ingestion for ten weeks significantly altered glucose and insulin levels in response to a glucose load.

In controls, blood glucose values at the different time points of the OGTT showed no significant differences at the beginning (0 weeks) and end (10 weeks) of the study (inter-action *p* value = 0.20) (Figure 2A). Likewise, serum insulin levels at the different time points of the OGTT exhibited no significant differences at the beginning and end of the study (interaction *p* value = 0.78) (Figure 2B). Volunteers drinking sucralose for ten weeks tended to display higher blood glucose values at all time points of the OGTT than those found basally at the beginning of the study. However, no significant differences were reached (interaction *p* value = 0.89) (Figure 2C). Conversely, participants consuming sucralose for ten weeks exhibited a significant 32 ± 3.5% increase in the serum insulin maximum peak at 30 min of the OGTT compared to that found basally at the beginning of the study (*p* < 0.001) (Figure 2D).

The area under the curve (AUC) of glucose and insulin confirmed what we found in OGTTs (Figure 3). At the beginning of the study, there were no differences between control and sucralose groups for the AUC of glucose (AUCG) (*p* = 0.99) (Figure 3A). Likewise, volunteers drinking water for ten weeks showed a similar AUCG than that found basally (*p* = 0.87) (Figure 3A). At the end of the study, we observed no differences between control and sucralose groups for the AUCG again (*p* = 0.27) (Figure 3A). Conversely, volunteers drinking sucralose for ten weeks exhibited a significant 8 ± 1.7% higher AUCG than that found basally at the beginning of the study (*p* = 0.02) (Figure 3A). The AUC of insulin (AUCI) showed no differences between control and sucralose groups at the beginning (*p* = 0.99) and end (*p* = 0.96) of the study (Figure 3B). In parallel, participants drinking water (*p* = 0.94) or sucralose (*p* = 0.83) for ten weeks exhibited similar AUCIs to those found basally at the beginning of the study (Figure 3B).

Besides examining the effect of sucralose on blood glucose and insulin, we also investigated whether this NAS affected the gut microbiome balance. For Actinobacteria, there were no differences between control and sucralose groups when analyzing the relative abundance of *Bifidobacterium longum* at the beginning and the end of the study (Figure 4A). We found a similar trend for Bacteroidetes, where the relative abundance of *Bacteroides uniformis* showed no differences between control and sucralose groups all along with the study (Figure 4B). For Firmicutes, volunteers drinking sucralose for ten weeks exhibited a significant 0.6-fold decrease in the relative abundance of *Lactobacillus acidophilus* compared to that found basally (*p* = 0.03) (Figure 4C). In parallel, sucralose intake for ten weeks induced significant 3 and 4-fold increases in the abundance of *Blautia coccoides* compared to the beginning of the study (*p* = 0.01) and the control group (*p* = 0.008) (Figure 4D).

Figure 5 illustrates statistical correlations of glucose and insulin values with changes in the relative abundance of *Lactobacillus acidophilus* and *Blautia coccoides* after water or sucralose consumption. In the control group, participants exhibited a positive correlation between AUCG and AUCI (*r* = 0.60, *p* = 0.01) (Figure 5A). AUCG also displayed a strong inverse correlation with *Lactobacillus acidophilus* (*r* = −0.56, *p* = 0.02) (Figure 5A). In turn, *Bifidobacterium longum* had a positive association with *Lactobacillus acidophilus* (*r* = 0.51, *p* = 0.03) (Figure 5A). Sucralose consumption considerably modified the intrinsic associations of gut microbiota with glucose and insulin levels. Volunteers in the sucralose group exhibited a positive association of AUCG with AUCI (*r*= 0.63, *p* < 0.01) (Figure 5B). Moreover, AUCG displayed strong inverse correlations with *Lactobacillus acidophilus* (*r* = −0.59, *p* < 0.02) and *Blautia coccoides* (*r* = −0.53, *p* < 0.02) (Figure 5B). Unexpectedly, AUCI exhibited a strong inverse relationship with *Lactobacillus acidophilus* (*r* = −0.55, *p* < 0.02), while *Lactobacillus acidophilus* showed a significant positive correlation with *Bacteroides uniformis* (*r* = 0.57, *p* < 0.02) (Figure 5B).

## 4. Discussion

Emerging evidence has linked the effect of sucralose on glucose and insulin blood levels with changes in the gut microbiome [12,13,14,15]. In line with this information, we confirmed that long-term sucralose consumption causes dysbiosis in healthy non-insulin resistant young adults, affecting the relative abundance of bacteria mainly belonging to the phylum Firmicutes.

In humans, the gut microbiome is composed of over 3 × 10^13^ bacteria [16], among which the phylum Firmicutes appear to play a central role in the metabolism of glucose and insulin [17]. Sucralose ingestion increases the relative amount of Firmicutes as hyperglycemia and insulin resistance rise in mice [3]. Patients with hyperinsulinemia and glucose intolerance show a higher intestinal abundance of Firmicutes, especially *Blautia coccoides* [18,19,20]. *Blautia coccoides* seem to counter the abnormal glucose and insulin homeostasis in patients with metabolic syndrome by producing butyrate that contributes to maintaining normal blood glucose levels [21,22]. Interestingly, we found that sucralose ingestion instigates an inverse relationship between AUCG and *Blautia coccoides*, which may decrease butyrate levels, inducing increased glucose release during the OGTT. Although plausible, this hypothesis still needs to be responded to by measuring butyrate levels to know whether changes in the intestinal abundance of *Blautia coccoides* may lead to variations in this metabolite and, consequently, an elevation in the AUCG.

*Lactobacillus acidophilus* is another essential member of the phylum Firmicutes with prominent actions in regulating insulin and glucose levels. A previous study in Sprague-Dawley rats demonstrated that low amounts of sucralose have antibacterial effects on *Lactobacillus acidophilus* [23]. In line with this evidence, our study indicates that sucralose intake for ten weeks decreases the relative abundance of *Lactobacillus acidophilus* in non-insulin-resistant young adults. Considering that consumption of *Lactobacillus acidophilus* improves carbohydrate metabolism in obese subjects and patients with type 2 diabetes [24,25], it is feasible that sucralose contributes to the abnormal insulin and glucose behavior by decreasing the *Lactobacillus acidophilus* population. Further clinical studies are needed to elucidate how sucralose reduces the intestinal amount of *Lactobacillus acidophilus*, contributing to altered glucose and insulin values in human beings.

The effects of sucralose consumption on the human gut microbiota are still a matter of debate. While previous clinical trials report no effects of sucralose on the relative abundance of intestinal bacteria [7,8], we found that consumption of this NAS increases the relative abundance of Firmicutes without affecting Bacteroidetes and Actinobacteria. We may attribute these controversial findings to variations in the experimental design, including duration of intervention and sucralose dosage. Ahmad and coworkers provided sucralose in two 2-week periods, incorporating a 4-week washout period [7]. Furthermore, according to the FDA, they used 40% of the ADI for sucralose [10,11]. Thomson and their colleagues performed a short-term clinical study for a week, using 229% of the ADI for this NAS [8]. Conversely, we opted for a long-term sucralose exposure for 10 weeks, using 48 mg sucralose per day representing less than 15% of the ADI for this NAS. Interestingly, the effects of sucralose consumption on insulin sensitivity appear in a bell-shaped dose-response form in humans [26], where the most potent effects present at much lower doses than those recommended in the ADI [12,13]. The microbiome composition may follow the same behavior, showing the most notable changes in response to low sucralose amounts. We think it is of great relevance to clarify whether sucralose acts differently depending on the size of the dose administered; conducting crossover clinical trials facilitates a comparison between the effects of low and high amounts of sucralose upon the gut microbiome. Additionally, we must consider other sources of variation in these results, such as genetic background, diet habits, and lifestyles, which may modify the effects of sucralose on the human intestinal microbiome.

The relevance of studying the effects of long-term sucralose consumption on the gut microbiome goes beyond insulin homeostasis and glycemic control [27,28]. A study in mice showed that sucralose ingestion for six weeks increases the relative abundance of bacteria belonging to the phylum Firmicutes, such as *Clostridium symbiosum* and *Peptostreptococcus anaerobius* [29]. Notably, sucralose-induced intestinal dysbiosis also appeared to aggravate azoxymethane (AOM)/dextran sulfate sodium (DSS)-induced colitis and colitis-associated colorectal cancer in these animals [29,30]. Likewise, sucralose ingestion resulted in gut dysbiosis and pronounced proteomic changes in the liver of mice, where most of the overexpressed proteins related to enhanced hepatic inflammation [31]. Thus, perturbations of intestinal microbiota, mediated by long-term sucralose consumption, not only seem associated with altered glucose and insulin homeostasis, but also with the promotion of inflammatory responses in the gut and liver. However, very recent studies also suggest that NAS, such as sucralose, ameliorates DSS-induced colitis symptoms by improving the intestinal barrier integrity and reducing intestinal inflammation via gut microbiota alteration in mice [32]. Therefore, we still need to conduct additional studies to clarify whether the adverse effects of sucralose on the gut microbiome could also increase the risk of developing inflammatory diseases, such as colitis and hepatitis in humans.

We are inclined to point out some strengths and limitations of this study. The adherence rate to water or sucralose consumption was acceptable by standing above 80% in our study population. Sample size (*n* = 20 per group) could be a limitation of the study. Regardless, it was a randomized clinical trial, which included daily follow-ups with volunteers for 10 weeks. Moreover, we should assess the effects of sucralose ingestion on gut microbiota composition by using other methodologies, such as Mi-Seq sequencing or 16S sequencing, that would allow us to properly characterize the changes in all of the bacterial species in the human gut. Finally, we should evaluate the impact of sucralose consumption on glucose and insulin blood values using a different method to OGTT that allows for better reproducibility.

## 5. Conclusions

This clinical trial demonstrates, for the first time, that consumption of 48 mg of sucralose every day for ten weeks induces gut dysbiosis by increasing *Blautia coccoides* and decreasing *Lactobacillus acidophilus* in healthy non-insulin-resistant young adults. As far as we know, this is one of the first long-term clinical trials showing that sucralose amounts, far lower than the suggested ADI, alter the balance of the gut microbiome, while also being associated with significant elevations in AUCG and serum insulin in response to glucose loads.

## Figures and Tables

**Figure 1 microorganisms-10-00434-f001:**
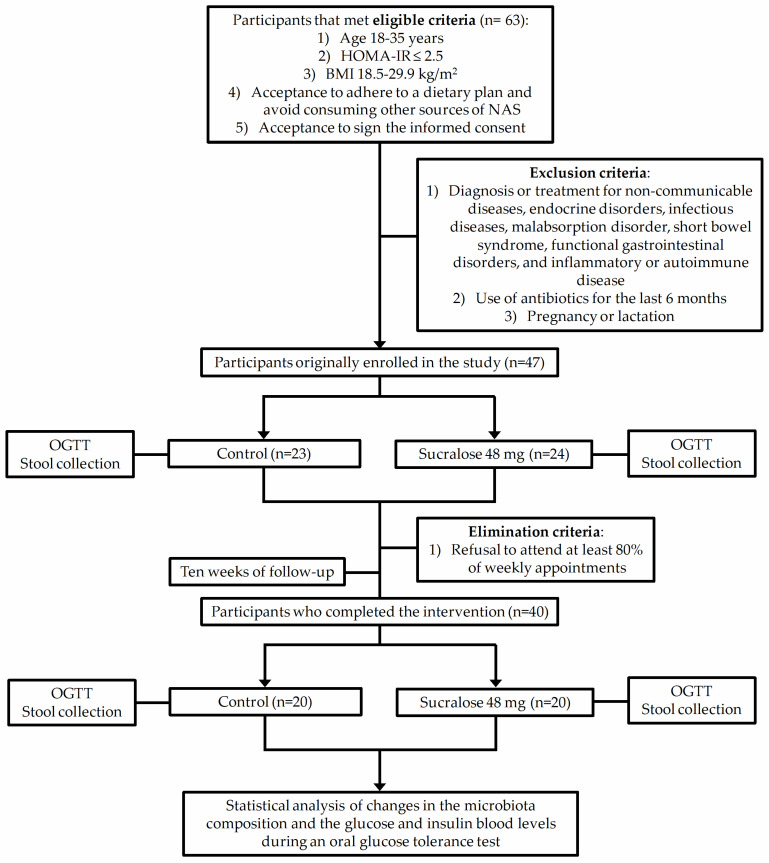
Schematic flow chart showing the selection process of participants enrolled in the study. Homeostasis model assessment of insulin resistance (HOMA-IR); body mass index (BMI); non-caloric artificial sweeteners (NAS); oral glucose tolerance test (OGTT).

**Figure 2 microorganisms-10-00434-f002:**
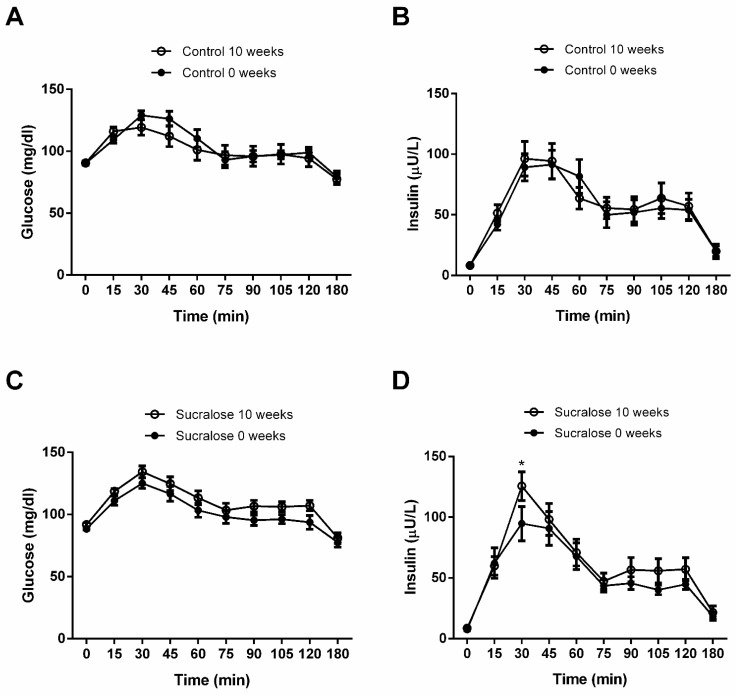
Blood levels of glucose and insulin during oral glucose tolerance test. (**A**) Volunteers receiving water for ten weeks showed no differences in blood glucose curves compared to the beginning of the study (0 weeks). (**B**) Participants receiving water for ten weeks exhibited no differences in serum insulin curves compared to the beginning of the study. (**C**) Volunteers drinking sucralose for ten weeks showed no differences in blood glucose curves compared to the beginning of the study. (**D**) Participants consuming sucralose for ten weeks exhibited a significant increase in serum insulin, with the maximum peak occurring 30 min after glucose load, compared to that found at the beginning of the study. We show control and sucralose groups at 0 weeks in closed circles, whereas both groups are shown in open circles at ten weeks. We express data as mean ± standard deviation. We estimated significant differences by two-tailed 2-way ANOVA with Bonferroni correction. Asterisks (*) indicate significant differences when *p* < 0.05.

**Figure 3 microorganisms-10-00434-f003:**
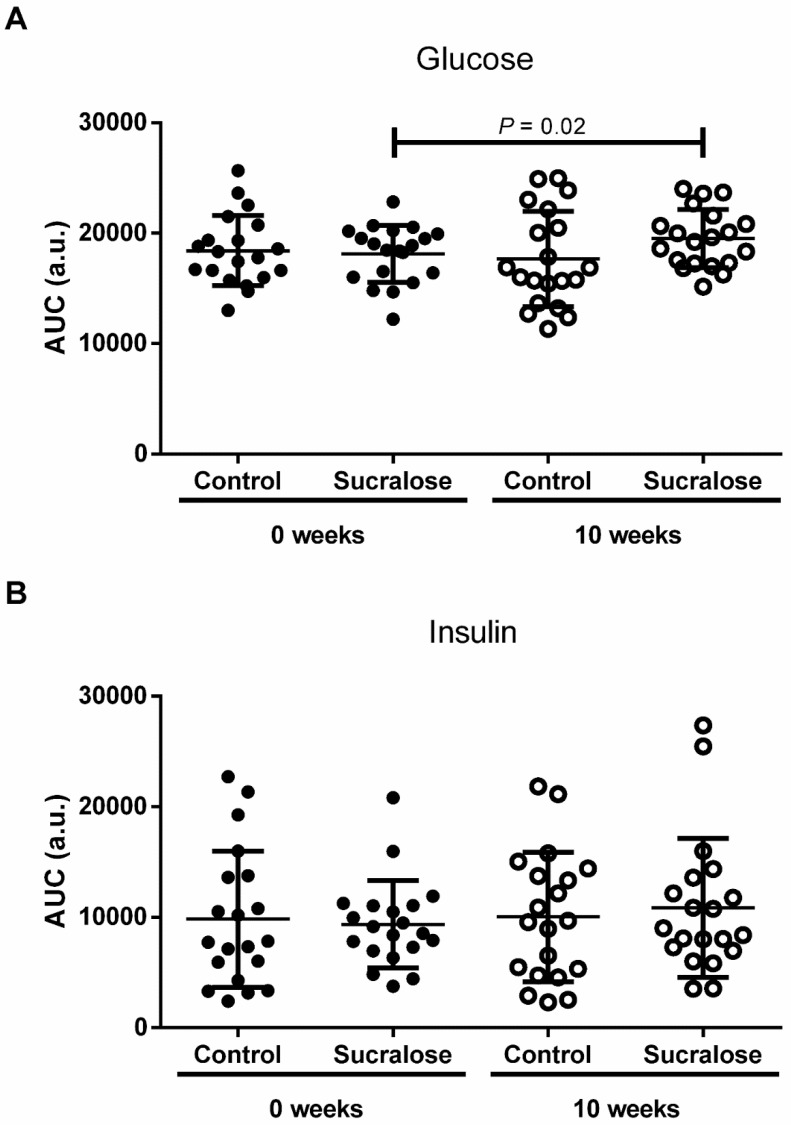
The area under the curve of glucose and insulin during the oral glucose tolerance test. (**A**) At the beginning and end of the study, there were no differences between control and sucralose groups for the AUC of glucose (AUCG). Volunteers drinking sucralose for ten weeks exhibited a higher AUCG than was found at the study’s commencement. (**B**) At the beginning and end of the study, there were no differences between control and sucralose groups for the AUC of insulin (AUCI). Volunteers drinking water or sucralose for ten weeks exhibited similar AUCIs to those found at the beginning of the study. We show control and sucralose groups at 0 weeks in closed circles, whereas both groups are shown in open circles at ten weeks. We express data as mean ± standard deviation. Depending on the group comparison, we estimated significant differences from the paired Student’s T-test or the unpaired Student’s T-test. We considered a difference as significant when *p* < 0.05. Area under the curve (AUC); arbitrary units (a.u.).

**Figure 4 microorganisms-10-00434-f004:**
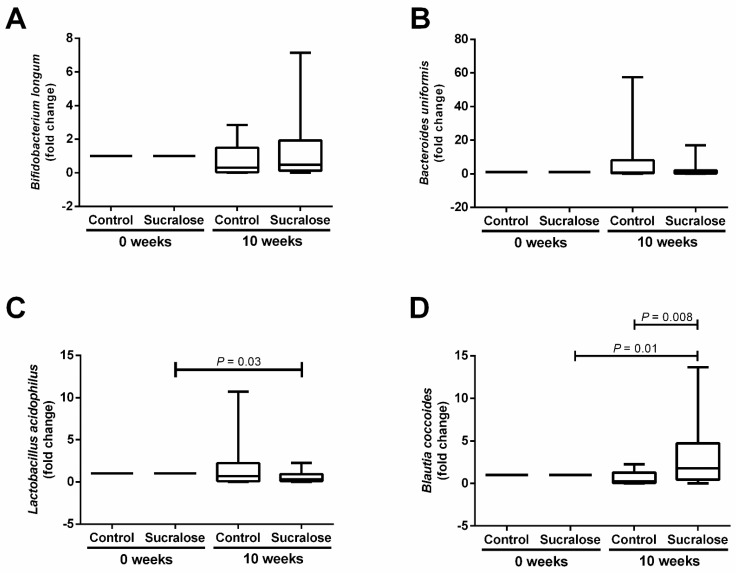
Effect of sucralose consumption on the relative abundance of Actinobacteria, Bacteroidetes, and Firmicutes. (**A**) For the phylum Actinobacteria, there were no differences between control or sucralose groups in the relative abundance of *Bifidobacterium longum* at the beginning (o weeks) or end (10 weeks) of the study. (**B**) For the phylum Bacteroidetes, there were no differences between control or sucralose groups in the relative abundance of *Bacteroides uniformis* at the beginning or end of the study. (**C**) For the phylum Firmicutes, participants drinking sucralose for ten weeks exhibited a significant decrease in the relative abundance of *Lactobacillus acidophilus* compared to the beginning of the study. (**D**) Volunteers consuming sucralose for ten weeks showed a significant increase in the relative abundance of *Blautia coccoides* compared to the beginning of the study and the control group. We express data as box plots. Depending on the group comparison, we estimated significant differences using the Wilcoxon matched-pairs signed-rank test or the Mann–Whitney U test. We calculated the fold change of the relative bacterial abundance by normalizing its corresponding value to 1 at the beginning of the study. We considered a difference as significant when *p* < 0.05.

**Figure 5 microorganisms-10-00434-f005:**
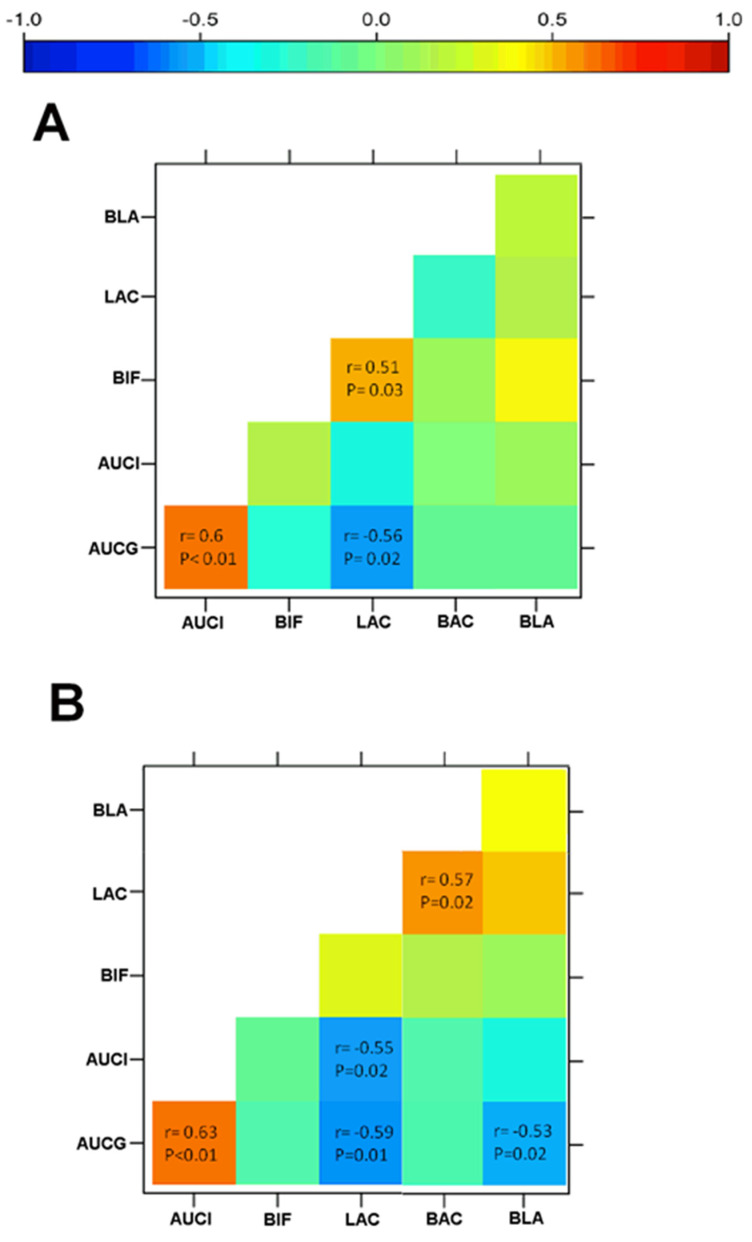
Heat maps showing correlations between the gut microbiome and glucose and insulin serum values. (**A**) Volunteers receiving water for ten weeks showed that AUCG positively correlated with AUCI and inversely associated with *Lactobacillus acidophilus* (LAC). *Bifidobacterium longum* (BIF) had a positive correlation with LAC. (**B**) Volunteers drinking sucralose every day for ten weeks showed a positive association of AUCG with AUCI. Moreover, AUCG inversely correlated with LAC and *Blautia coccoides* (BLA). AUCI exhibited a strong inverse relationship with LAC, while LAC showed a significant positive correlation with *Bacteroides uniformis* (BAC). We calculated coefficients (*r*) and *p* values by using the Spearman’s correlation model. Inverse correlations are stronger as blue color increases, whereas positive correlations are stronger as red color increases. We considered a correlation as significant when *p* < 0.05. AUCG, area under the curve of glucose; AUCI, area under the curve of insulin.

**Table 1 microorganisms-10-00434-t001:** Demographic, anthropometric, and biochemical characteristics of the participants at the beginning and end of the study.

Parameter	Control (*n* = 20)		*p* Value	Sucralose (*n* = 20)		*p* Value	*p* Value ^ac^	*p* Value ^bd^
	Basal ^a^	10 Weeks ^b^		Basal ^c^	10 Weeks ^d^			
Gender (w/m)	12/8	12/8	1.0	14/6	14/6	1.0	0.7	0.7
Age (years)	22.7 ± 3.8	22.8 ± 3.7	0.8	22.9 ± 3.4	23 ± 3.3	0.7	0.5	0.4
Physical activity (yes/no)	8/12	6/14	0.7	7/13	9/11	0.7	1.0	0.5
Weight (kg)	66.7 ± 13.5	66.5 ± 13.6	0.2	63.9 ± 11.9	64.2 ± 12.3	0.2	0.3	0.3
BMI (kg/m^2^)	24.9 ± 4.5	24.7 ± 4.6	0.1	24.3 ± 2.6	23.5 ± 4.9	0.4	0.9	0.9
Waist circumference (cm)	80.8 ± 10.6	81 ± 10.7	0.6	80.4 ± 8	79.7 ± 7.9	0.7	0.7	0.7
Hip circumference (cm)	99.5 ± 9.9	99.6 ± 10.6	0.9	98.7 ± 6.2	98.9 ± 6.7	0.6	0.8	0.9
Fat (%)	36.3 ± 8.4	36.3 ± 8.2	0.8	37.7 ± 4.6	37.9 ± 5.3	0.8	0.6	0.7
Total body water (%)	46 ± 6	45.8 ± 5.9	0.7	44.5 ± 4	44.6 ± 4.2	0.7	0.5	0.5
Lean dry mass (%)	17.6 ± 2	17.7 ± 2.6	0.2	17.6 ± 1.4	17.6 ± 1.5	0.3	0.7	0.4
SBP (mmHg)	112 ± 9.4	112 ± 7	0.5	110 ± 10.9	111 ± 8.7	0.5	0.6	0.7
DBP (mmHg)	72.7 ± 6.5	74 ± 4.7	0.1	72.7 ± 6.7	72.5 ± 8.5	0.8	0.7	0.4
Blood glucose (mg/dL)	87 ± 5.4	88 ± 7.2	0.2	89.1 ± 5.5	90.2 ± 4.3	0.2	0.2	0.3
HbA1c (%)	5.2 ± 0.2	5.2 ± 0.3	0.6	5.2 ± 0.2	5.2 ± 0.2	0.8	0.7	0.9
Serum insulin (mU/L)	7.7 ± 2.8	8.1 ± 3.5	0.6	7.7 ± 2.7	8.3 ± 4.5	0.6	0.8	0.8
HOMA-IR (a.u.)	1.6 ± 0.5	1.7 ± 0.8	0.8	1.7 ± 0.6	1.7 ± 0.7	0.9	0.8	0.7
Triglycerides (mg/dL)	80.6 ± 33.3	98.2 ± 45.3	0.08	102.1 ± 64.3	97 ± 60.8	0.2	0.06	0.8
Total cholesterol (mg/dL)	159.4 ± 29.2	165 ± 39.6	0.5	163.5 ± 28.1	161.6 ± 31.5	0.4	0.5	0.7
LDL (mg/dL)	93.4 ± 25.5	99.1 ± 34.5	0.3	93.2 ± 20.1	93.9 ± 21.4	0.7	0.8	0.6
HDL (mg/dL)	44.3 ± 11.3	46.2 ± 9.8	0.3	41.8 ± 10.1	44.1 ± 11.2	0.1	0.4	0.5
Serum creatinine (mg/dL)	0.7 ± 0.1	0.8 ± 1	0.5	0.7 ± 0.1	0.7 ± 0.2	0.2	0.6	0.6
Lipids (g/day)	70.8 ± 39.4	68.5 ± 38.2	0.1	69.3 ± 45.1	66.4 ± 40.7	0.4	0.5	0.1
Carbohydrates (g/day)	267.3 ± 101.6	252 ± 11.7	0.1	285.2 ± 116.3	249 ± 104.5	0.4	0.2	0.2
Protein (g/day)	105.7 ± 70.8	107.5 ± 68.1	0.5	99.4 ± 65.2	101.2 ± 69.3	0.8	0.8	0.8
Energy (kcal/day)	2090 ± 941	1857 ± 671	0.4	1994 ± 792	2167 ± 802	0.2	0.9	0.2

We express data as mean ± standard deviation except for gender and physical activity that we show as absolute values. Depending on the group comparison, we estimated significant differences using the Wilcoxon matched-pairs signed-rank test or the Mann–Whitney U test except for gender and physical activity that we assessed by the chi-squared test. We considered a difference as significant when *p* < 0.05. ^a^ Parameters at the beginning of the study in the control group, ^b^ parameters at the end of the study in the control group, ^c^ parameters at the beginning of the study in the sucralose group, ^d^ parameters at the end of the study in the sucralose group. Abbreviations: w, women; m, men; BMI, body mass index; SBP, systolic blood pressure; DBP, diastolic blood pressure; HbA1c, glycated hemoglobin; HOMA-IR, homeostatic model assessment of insulin resistance; LDL, low-density lipoprotein; HDL, high-density lipoprotein; a.u., arbitrary units; kcal, kilocalories.

## Data Availability

The data presented in this study are available upon request.

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
