# Peer review of "Ten-Week Sucralose Consumption Induces Gut Dysbiosis and Altered Glucose and Insulin Levels in Healthy Young Adults"

_microorganisms, 2022, doi:10.3390/microorganisms10020434_

Round 1
Reviewer 1 Report
A very interesting article. I think it will bring a lot to the world of science when it comes to understanding the effects of substances on the microbiota.
I have some suggestions:
- line 152: I would not describe this intervention as a "treatment" because that is not the goal
- line 159: please make Table 1 in bold.
-figure 2d - please explain what this is *
- figures 5A and B not very clear
- please write exactly what methods were used to monitor the diet. Was physical activity monitored?
-improve the bibliography according to the guidelines for authors
- the literature contains only 26 items. Do you see the possibility of extending the literature?
Author Response
QUERY 1 (Q1). A very interesting article. I think it will bring a lot to the world of science when it comes to understanding the effects of substances on the microbiota. I have some suggestions:
REPLY 1 (R1). We sincerely thank you for your kind comments on our work.
Q2. line 152: I would not describe this intervention as a "treatment" because that is not the goal.
R2. We deleted the term mentioned above and rephrased the sentence following the Reviewer's observation. Please find this change marked with yellow on page 5.
Q3. line 159: please make Table 1 in bold.
R3. We bolded Table 1 as suggested. Please find this change marked with green on page 5.
Q4. figure 2d - please explain what this is *
R4. Please see the description of asterisks in Figure 2D, marked with yellow on page 7.
Q5. figures 5A and B not very clear
R5. We replaced Figure 5 using another one with increased quality. Furthermore, we rephrased the first part of the paragraph explaining findings in Figure 5 to clarify. Please find these changes marked yellow on page 10 and Figure 5 on page 11.
Q6. please write exactly what methods were used to monitor the diet. Was physical activity monitored?
R6. We weekly monitored that all participants followed a balanced diet and calculated the amount of ingested protein, carbohydrates, and lipids via phone call interview. Please see this information marked with yellow on page 2. We also registered each participant’s physical activity levels weekly via phone interview and found no significant changes between volunteers drinking water or sucralose. Please see this information marked blue on pages 2 and 5 and Table 1.
Q7. improve the bibliography according to the guidelines for authors
R7. We formatted the reference list according to the author’s guidelines. Please see these changes in the reference list on pages 14 and 15.
Q8. the literature contains only 26 items. Do you see the possibility of extending the literature?
R8. We discussed and incorporated six new references to the reference list following your observation. Please see these changes marked yellow on page 15.
We sincerely thank you for all your valuable comments and suggestions, which have indubitably improved the last version of the manuscript.

Reviewer 2 Report
Re: Manuscript ID: microorganisms-1577529.
This is a well written research article dealing with the role of sucralose in gut dysbiosis, with a particular attention to glucose and insulin metabolism in healthy young adults. The authors demonstrate for the first time that consumption of sucralose can induce gut dysbiosis, leading to alterations in glucose and insulin levels during an oral glucose load. The work does not seem relevant in this field and the authors outlined the limitations of their study. Some changes are suggested to improve the paper.
Points of criticism
When the participants to the study had their breakfast with respect to the administration of sucralose/water?
Did the participants observe a particular diet regimen during the study? Did they take probiotics or prebiotics? Please, describe the dietary plan mentioned in the article.
Did the participants practice physical activity during the study?
I suggest the authors to consider and discuss the following articles, also with respect to colitis and cancer:
Ahmad SY, Friel JK, Mackay DS. Effect of sucralose and aspartame on glucose metabolism and gut hormones. Nutr Rev. 2020 Sep 1;78(9):725-746. doi: 10.1093/nutrit/nuz099.
Ahmad SY, Azad MB, Friel J, MacKay D. Recent evidence for the effects of nonnutritive sweeteners on glycaemic control. Curr Opin Clin Nutr Metab Care. 2019 Jul;22(4):278-283. doi: 10.1097/MCO.0000000000000566.
Dai X, Guo Z, Chen D, Li L, Song X, Liu T, Jin G, Li Y, Liu Y, Ajiguli A, Yang C, Wang B, Cao H. Maternal sucralose intake alters gut microbiota of offspring and exacerbates hepatic steatosis in adulthood. Gut Microbes. 2020 Jul 3;11(4):1043-1063. doi: 10.1080/19490976.2020.1738187.
Li X, Liu Y, Wang Y, Li X, Liu X, Guo M, Tan Y, Qin X, Wang X, Jiang M. Sucralose Promotes Colitis-Associated Colorectal Cancer Risk in a Murine Model Along With Changes in Microbiota. Front Oncol. 2020 Jun 3;10:710. doi: 10.3389/fonc.2020.00710.
Guo M, Liu X, Tan Y, Kang F, Zhu X, Fan X, Wang C, Wang R, Liu Y, Qin X, Jiang M, Wang X. Sucralose enhances the susceptibility to dextran sulfate sodium (DSS) induced colitis in mice with changes in gut microbiota. Food Funct. 2021 Oct 4;12(19):9380-9390. doi: 10.1039/d1fo01351c.
Liu CW, Chi L, Tu P, Xue J, Ru H, Lu K. Quantitative proteomics reveals systematic dysregulations of liver protein metabolism in sucralose-treated mice. J Proteomics. 2019 Mar 30;196:1-10. doi: 10.1016/j.jprot.2019.01.011.
Zhang X, Gu J, Zhao C, Hu Y, Zhang B, Wang J, Lv H, Ji X, Wang S. Sweeteners Maintain Epithelial Barrier Function Through the miR-15b/RECK/MMP-9 Axis, Remodel Microbial Homeostasis, and Attenuate Dextran Sodium Sulfate-Induced Colitis in Mice. J Agric Food Chem. 2022 Jan 12;70(1):171-183. doi: 10.1021/acs.jafc.1c06788.
Author Response
QUERY 1 (Q1). This is a well written research article dealing with the role of sucralose in gut dysbiosis, with a particular attention to glucose and insulin metabolism in healthy young adults. The authors demonstrate for the first time that consumption of sucralose can induce gut dysbiosis, leading to alterations in glucose and insulin levels during an oral glucose load. The work does not seem relevant in this field and the authors outlined the limitations of their study. Some changes are suggested to improve the paper. Points of criticism:
REPLY 1 (R1). We thank you for your kind comments on our work.
Q2. When the participants to the study had their breakfast with respect to the administration of sucralose/water?
R2. All participants reported having breakfast between 7 and 9 am, and 60 ml sucralose or water were drunk ten minutes before breakfast. We added this information to the Material and Method section for clarification. Please find this change marked with red on page 3.
Q3. Did the participants observe a particular diet regimen during the study? Did they take probiotics or prebiotics? Please, describe the dietary plan mentioned in the article.
R3. Participants did not observe a particular diet regimen during the study. Still, according to their physical activity level, we advised each participant to follow a balanced diet, including vegetables, fruits, grains, proteins, and dairy. We also monitored weekly the amount of protein, carbohydrates, and lipids ingested by each participant and found no significant differences between sucralose and water groups. All participants did not take either probiotics or prebiotics during the study. We think your observation is of great relevance, and to avoid misunderstandings, we included a paragraph describing all this information in the Material and Method section. Please see this information marked with yellow on page 2.
Q4. Did the participants practice physical activity during the study?
R4. We registered each participant's physical activity levels weekly that mainly consisted of running for no more than forty minutes, two or three days per week. There were no significant differences between sucralose and control groups for the physical activity level. Following your observation, we added this information for clarification. Please see this information marked blue on pages 2 and 5 and Table 1.
Q5. I suggest the authors to consider and discuss the following articles, also with respect to colitis and cancer:
-Ahmad SY, Friel JK, Mackay DS. Effect of sucralose and aspartame on glucose metabolism and gut hormones. Nutr Rev. 2020 Sep 1;78(9):725-746. doi: 10.1093/nutrit/nuz099.
-Ahmad SY, Azad MB, Friel J, MacKay D. Recent evidence for the effects of nonnutritive sweeteners on glycaemic control. Curr Opin Clin Nutr Metab Care. 2019 Jul;22(4):278-283. doi: 10.1097/MCO.0000000000000566.
-Dai X, Guo Z, Chen D, Li L, Song X, Liu T, Jin G, Li Y, Liu Y, Ajiguli A, Yang C, Wang B, Cao H. Maternal sucralose intake alters gut microbiota of offspring and exacerbates hepatic steatosis in adulthood. Gut Microbes. 2020 Jul 3;11(4):1043-1063. doi: 10.1080/19490976.2020.1738187.
-Li X, Liu Y, Wang Y, Li X, Liu X, Guo M, Tan Y, Qin X, Wang X, Jiang M. Sucralose Promotes Colitis-Associated Colorectal Cancer Risk in a Murine Model Along With Changes in Microbiota. Front Oncol. 2020 Jun 3;10:710. doi: 10.3389/fonc.2020.00710.
-Guo M, Liu X, Tan Y, Kang F, Zhu X, Fan X, Wang C, Wang R, Liu Y, Qin X, Jiang M, Wang X. Sucralose enhances the susceptibility to dextran sulfate sodium (DSS) induced colitis in mice with changes in gut microbiota. Food Funct. 2021 Oct 4;12(19):9380-9390. doi: 10.1039/d1fo01351c.
-Liu CW, Chi L, Tu P, Xue J, Ru H, Lu K. Quantitative proteomics reveals systematic dysregulations of liver protein metabolism in sucralose-treated mice. J Proteomics. 2019 Mar 30;196:1-10. doi: 10.1016/j.jprot.2019.01.011.
-Zhang X, Gu J, Zhao C, Hu Y, Zhang B, Wang J, Lv H, Ji X, Wang S. Sweeteners Maintain Epithelial Barrier Function Through the miR-15b/RECK/MMP-9 Axis, Remodel Microbial Homeostasis, and Attenuate Dextran Sodium Sulfate-Induced Colitis in Mice. J Agric Food Chem. 2022 Jan 12;70(1):171-183. doi: 10.1021/acs.jafc.1c06788.
R5. Following your suggestion, we incorporated the abovementioned articles in the discussion section. Then, we discussed the potential effects of long-term sucralose ingestion on the gut microbiota concerning the occurrence of colitis, colorectal cancer, and liver inflammation. Please see this change marked with yellow on pages 13 and 15.
We sincerely thank you for all your valuable comments and observations, which have indubitably improved the last version of the manuscript.

This manuscript is a resubmission of an earlier submission. The following is a list of the peer review reports and author responses from that submission.
Round 1
Reviewer 1 Report
In the manuscript ID-microorganisms-1294176 titled “Ten-week sucralose consumption induces gut dysbiosis and altered homeostasis of glucose and insulin in healthy young adults” by Lucia Angelica Méndez-García and colleagues. They have reported results from a double-blind placebo-controlled clinical trial, in which volunteers randomly received water as a placebo (n=20) or 48 mg sucralose (n=20) every day for 10 weeks effects of sucralose in gut microbiota. Sucralose intake altered the abundance of Firmicutes without affecting Actinobacteria and Bacteroidetes. Sucralose consumption also increased serum insulin and the area under the curve of glucose concerning placebo. Sucralose ingestion induces gut dysbiosis and altered homeostasis of insulin and glucose in humans. I have few concerns regarding the present manuscript.
-The topic that the authors have investigated is interesting and novel in the field. In the introduction, they need new information for the effects of NAS in the gut microbiota and check the names of the different phylum and genera, maybe in italics, and also the style for the references.
-Information about ADI is missing in the entire document
-If the authors have mentioned the CONSORT guidelines, they need to add a flow-chart diagram about enrollment, intervention, …among others.
-Why the authors decide for quantitative PCR instead Mi-Seq sequencing or other similar, they have the DNA, and maybe a 16S sequencing is a better approach than PCR
-The authors have validated the different primers/probes in their analyzed population.
-What is the control in the PCR, and also, when we have a lot of probes we need to follow the MIQE that establish at least three controls in the whole experiment
-Seems that the ingestion of sucralose has effects on the gut microbiota, however, the authors failed to obtain some data from the diet, questionnaires about dietary variables are really important in these studies.
-Thank you for add limitations, for me the most important is the technique and also the missing dietary variables.
Author Response
Reviewer #1
In the manuscript ID-microorganisms-1294176 titled “Ten-week sucralose consumption induces gut dysbiosis and altered homeostasis of glucose and insulin in healthy young adults” by Lucia Angelica Méndez-García and colleagues. They have reported results from a double-blind placebo-controlled clinical trial, in which volunteers randomly received water as a placebo (n=20) or 48 mg sucralose (n=20) every day for 10 weeks effects of sucralose in gut microbiota. Sucralose intake altered the abundance of Firmicutes without affecting Actinobacteria and Bacteroidetes. Sucralose consumption also increased serum insulin and the area under the curve of glucose concerning placebo. Sucralose ingestion induces gut dysbiosis and altered homeostasis of insulin and glucose in humans. I have few concerns regarding the present manuscript.
Reply (R)
We thanks to the Reviewer for her/his criticism.
Query (Q) 1
-The topic that the authors have investigated is interesting and novel in the field. In the introduction, they need new information for the effects of NAS in the gut microbiota and check the names of the different phylum and genera, maybe in italics, and also the style for the references.
R1
Thank you for your very nice comments on our work. Following your observation, we added more information regarding the effect of NAS, especially sucralose, on the gut microbiome. Please find this information marked with yellow color at page 2.
We also corrected the names of phylum (phyla), order (orders), genus (genera), and species, having special emphasis on writing the genus and species names in italics all along the text. Please see these changes marked with fuchsia color at pages 1, 2, 4, 9, 11, 13, 14, and 15, and in Figure 4 and its figure legend.
Moreover, we corrected the reference style, according to the instructions for authors. Please find these changes in the Reference section at pages 15-18.
Q2
-Information about ADI is missing in the entire document
R2
Following your observation, we added information regarding the Acceptable Daily Intake (ADI) for sucralose, according to the Food and Drug Administration (FDA) and the Joint Food and Agriculture Organization (FAO)/World Health Organization Expert Committee on Food Additives (JECFA). Please find this information marked with green color at pages 3 and 14.
Q3
-If the authors have mentioned the CONSORT guidelines, they need to add a flow-chart diagram about enrollment, intervention, …among others.
R3
We totally agree with the Reviewer’s observation. For this reason, we added a flow chart illustrating the selection process of participants enrolled in the study, having special emphasis on the intervention, follow-up, and experimentation. Please find this information in the new Figure 1, marked with red color at pages 5 and page 6.
Q4
-Why the authors decide for quantitative PCR instead Mi-Seq sequencing or other similar, they have the DNA, and maybe a 16S sequencing is a better approach than PCR
R4
We thank to the Reviewer for her/his criticism. The main goal of our study was to examine the effect of 48 mg sucralose ingestion on the relative abundance of 4 bacterial orders belonging to Actinobacteria (Bifidobacteriales), Bacteroidetes (Bacteroidales), and Firmicutes (Lactobacillales and Eubacteriales) that clearly differ between subjects with and without insulin resistance, glucose intolerance, and/or hyperglycemia, as reported by Nadja Larsen and colleagues (Larsen N et al., PLoS One. 5(2):e9085). For this reason, we think qPCR is enough to quantify and compare the relative abundance of the abovementioned bacteria at the beginning and at the end of the intervention. We concur with the Reviewer that Mi-Seq sequencing or other similar allows characterizing bacterial species through whole genome sequencing; however, our goal was only limited to quantify and compare the relative abundance of 4 well identified bacterial orders with potential implications in the homeostasis of glucose and insulin. Furthermore, qPCR is much cheaper than Mi-Seq sequencing or 16S sequencing, a factor that is sadly important to consider when conducting research projects in our country.
However, we think the Reviewer’s comment is of great importance and for this reason, we added a sentence pointing out this limitation of the study. Please find this information marked with red color at page 15.
Q5
-The authors have validated the different primers/probes in their analyzed population.
R5
We validated the primers using specific isolates of Bifidobacteriales, Bacteroidales, Lactobacillales, and Eubacteriales. These bacterial isolates have been purified from patients attending to the Department of Internal Medicine and the Department of Gastroenterology of the General Hospital of Mexico and are available as part of the medical record of the same institution. We think your comment is of great relevance and for this reason, we added this information at the Material and Methods section. Please find this information marked with grey color at page 4.
Q6
-What is the control in the PCR, and also, when we have a lot of probes we need to follow the MIQE that establish at least three controls in the whole experiment.
R6
We used a Pan-bacteria universal primer as control. Moreover, qPCR data resulted from triplicates, according to the Minimum Information for Publication of Quantitative Real-Time PCR Experiments guidelines. Following your observation, we added this information to the Material and Methods section. Please find this information marked with yellow color at page 4.
Q7
-Seems that the ingestion of sucralose has effects on the gut microbiota, however, the authors failed to obtain some data from the diet, questionnaires about dietary variables are really important in these studies.
R7
We totally concur with the Reviewer’s observation. In fact, all participants enrolled in the study agreed to adhere strictly to a dietary plan that was individually standardized by the Nutrition Department of the General Hospital of Mexico and monitored weekly for registering daily consumption of lipids, carbohydrates, and proteins, among other important dietary variables. Originally, we thought those data were irrelevant. However, following the Reviewer’s observation we added some important dietary variables that might influence the gut microbiota composition such as daily consumption (g/day) of lipids, carbohydrates, and proteins in both control and sucralose groups. It is worth mentioning that consumption of lipids, carbohydrates, and proteins did not differ in both placebo and sucralose groups at the beginning and at the end of the study. Please find this information marked with turquoise color at pages 3, 6, and 8, and in Table 1.
Q8
-Thank you for add limitations, for me the most important is the technique and also the missing dietary variables.
R8
We sincerely thanks to the Reviewer for her/his accurate observations. We have tried to reply to all your concerns in a detailed point-by-point fashion, having special emphasis on providing the rationale behind using qPCR to assess changes in the relative abundance of intestinal bacteria, and more detailed information concerning important dietary variables. We certainly think that your comments and observations have improved the last version of the manuscript. Thank you.

Reviewer 2 Report
This study by L A. Mendez-Garcia et al is a double-blind placebo-controlled clinical trial. Non -insulin resistant young adults were subjected to consumption of 48 mg sucralose every day during 10 weeks, and the objective of the study was to assess the effects on microbiota, and on serum glucose and insulin.
I have some methodological concerns:
-Sucralose consumption had very weak effects on glucose and insulin secretion, and led to no significant difference on OGTT, which is not in accordance with other publications such as Lertrit et al. Nutrition 2018, or Romo-Romo et al., The American journal of clinical nutrition, 2018. The same research team has published a very similar study in non-insulin resistant healthy young adults showing more effects of 48 mg sucralose consumption than the submitted study (Bueno-Hernandez et al, Nutrition Journal 2020 Chronic sucralose consumption induces elevation of serum insulin in young healthy adults: a randomized, double blind, controlled trial). One would have expected at least a discussion about these discrepancies.
- The main aim of the study is the evaluation of the effects on gut microbiota. One can regret that the authors have chosen to assess microbiota by real-time PCR. The use of quantitative PCR with DDCT normalization only allows to compare the relative abundance of targeted bacteria between 2 experimental groups. Its use to assess whole microbiome profile as in figure 4 is not allowed. Moreover, I checked the first set of primers presented in the submitted manuscript (lines 150-151): it indeed targets both Clostridium coccoides and Eubacteria rectale (the choice of this set of primers in addition to another specifically targeting Clostridium coccoides (lines 153-154) is also questionable), but also Faecalicatena orotica and Blautia argi by instance. One can have serious doubts about the results obtained. Moreover the authors observed numerous problematic significant variations of bacteria DNA expression between control subjects at 0 weeks and Sucralose subjects at 0 weeks. Finally there is extrapolation or confusion on bacterial taxa. There are no data allowing to conclude about the abundance of bacterial phyla such as Firmicutes, Bacteroidetes or Actinobacteria (legend of figure 3, abstract lines 32-33)
-The novelty of the study is also questionable: see The Effects of Non-Nutritive Artificial Sweeteners, Aspartame and Sucralose, on the Gut Microbiome in Healthy Adults: Secondary Outcomes of a Randomized Double-Blinded Crossover Clinical Trial by Samar Y. Ahmad, James Friel and Dylan Mackay . Nutrients 2020, 12(11), 3408; https://doi.org/10.3390/nu12113408
Author Response
Reviewer #2
This study by L A. Mendez-Garcia et al is a double-blind placebo-controlled clinical trial. Non -insulin resistant young adults were subjected to consumption of 48 mg sucralose every day during 10 weeks, and the objective of the study was to assess the effects on microbiota, and on serum glucose and insulin.
I have some methodological concerns:
Reply (R)
We thanks to the Reviewer for her/his criticism.
Query (Q) 1
-Sucralose consumption had very weak effects on glucose and insulin secretion, and led to no significant difference on OGTT, which is not in accordance with other publications such as Lertrit et al. Nutrition 2018, or Romo-Romo et al., The American journal of clinical nutrition, 2018. The same research team has published a very similar study in non-insulin resistant healthy young adults showing more effects of 48 mg sucralose consumption than the submitted study (Bueno-Hernandez et al, Nutrition Journal 2020 Chronic sucralose consumption induces elevation of serum insulin in young healthy adults: a randomized, double blind, controlled trial). One would have expected at least a discussion about these discrepancies.
R1
We totally concur with the Reviewer. In fact, following the accurate Reviewer´s observation, we added several discussion paragraphs about these apparently controversial findings. Please find this information marked with yellow color at pages 13 and 14.
Q2
- The main aim of the study is the evaluation of the effects on gut microbiota. One can regret that the authors have chosen to assess microbiota by real-time PCR. The use of quantitative PCR with DDCT normalization only allows to compare the relative abundance of targeted bacteria between 2 experimental groups. Its use to assess whole microbiome profile as in figure 4 is not allowed.
R2
We thank to the Reviewer for her/his criticism. The main goal of our study was to examine the effect of 48 mg sucralose ingestion on the relative abundance of specific bacterial groups that clearly differ between subjects with and without insulin resistance, glucose intolerance, and/or hyperglycemia, as reported by Nadja Larsen and colleagues (Larsen N et al., PLoS One. 5(2):e9085). For this reason, we think qPCR is enough to quantify and compare the relative abundance of the abovementioned bacteria at the beginning and at the end of the intervention. We concur with the Reviewer that more robust methodologies such as Mi-Seq sequencing or other similar allow characterizing even bacterial species through whole genome sequencing; however, our goal was only limited to quantify and compare the relative abundance of previously well identified bacterial orders with potential implications in the homeostasis of glucose and insulin, without trying to characterize the whole microbiome profile.
However, we think your comment is of great importance and for this reason, we added a sentence pointing out this limitation of the study. Please find this information marked with red color at page 15.
Additionally, following your observation, Figure 4 was removed from the manuscript.
Q3a
Moreover, I checked the first set of primers presented in the submitted manuscript (lines 150-151): it indeed targets both Clostridium coccoides and Eubacteria rectale (the choice of this set of primers in addition to another specifically targeting Clostridium coccoides (lines 153-154) is also questionable), but also Faecalicatena orotica and Blautia argi by instance. One can have serious doubts about the results obtained.
R3a
We sincerely apologize for the misunderstanding. The main goal of the study was to examine the effect of sucralose consumption on the relative abundance of bacterial groups belonging to the phyla Actinobacteria, Bacteroidetes, and Firmicutes, which have been previously identified as potential contributors to metabolic dysfunction. We did not want to explore the whole intestinal microbiome or specific bacterial species and for this reason, we apologize for this misunderstanding. Following the Reviewer´s suggestion and to avoid data misinterpretation, we reformatted the manuscript by analyzing changes in the relative abundance of the orders Bifidobacteriales, Bacteroidales, Lactobacillales, and Eubacteriales that belong to phyla Actinobacteria, Bacteroidetes, and Firmicutes and include genera such as Bifidobacterium (Actinobacteria-Bifidobacteriales), Bacteroides (Bacteroidetes-Bacteroidales), Prevotella (Bacteroidetes-Bacteroidales), Roseburia (Firmicutes-Eubacteriales), Clostridium (Firmicutes-Eubacteriales), Eubacterium (Firmicutes-Eubacteriales), Blautia (Firmicutes-Eubacteriales), Faecalicatena (Firmicutes-Eubacteriales), and Lactobacillus (Firmicutes-Lactobacillales), among others. In this way, results are now clearer and more informative. Please see these changes marked with fuchsia color at pages 1, 2, 4, 9-15, and in new Figures 4 and 5, and their figure legends.
We sincerely thank you for your accurate observations that have indubitably improved the clarity of the manuscript.
Q3b
Moreover the authors observed numerous problematic significant variations of bacteria DNA expression between control subjects at 0 weeks and Sucralose subjects at 0 weeks.
R3b
We respectfully want to clear up that we did not find any important variation of bacterial DNA expression between control and sucralose groups at the beginning of the study (0w). In order to avoid any bias when compared the long-term effect of placebo or sucralose ingestion on gut microbiota, we decided to normalize bacterial DNA expression in each subject to 1 (0w). After ten weeks (10w), we quantified bacterial DNA expression in each participant again, and calculated its variation with respect to the beginning of the study as fold change. As it can be seen, there were no variations of bacterial DNA expression between the studied groups at the beginning of the study.
Q3c
Finally there is extrapolation or confusion on bacterial taxa. There are no data allowing to conclude about the abundance of bacterial phyla such as Firmicutes, Bacteroidetes or Actinobacteria (legend of figure 3, abstract lines 32-33).
R3c
We sincerely apologize for the misunderstanding in naming bacterial taxa. We have meticulously corrected the names of phylum (phyla), order (orders), genus (genera), and species, having special emphasis on writing the genus and species names in italics all along the text. Please see these changes marked with fuchsia color at pages 1, 2, 4, 9, 11, 13, 14, and 15, and in Figure 4 and its figure legend.
As mentioned above, the main goal of the study was to examine 4 representative orders of the phyla Actinobacteria, Bacteroidetes, and Firmicutes but not the whole bacterial species in each phylum. For this reason, we respectfully believe that our data allow illustrating the effect of long-term sucralose consumption on the relative abundance of Bifidobacteriales, Bacteroidales, Lactobacillales, and Eubacteriales, 4 bacterial orders that have been shown to play key roles in the metabolism of glucose and insulin.
Q4
-The novelty of the study is also questionable: see The Effects of Non-Nutritive Artificial Sweeteners, Aspartame and Sucralose, on the Gut Microbiome in Healthy Adults: Secondary Outcomes of a Randomized Double-Blinded Crossover Clinical Trial by Samar Y. Ahmad, James Friel and Dylan Mackay . Nutrients 2020, 12(11), 3408; https://doi.org/10.3390/nu12113408
R4
We thank to the Reviewer for her/his criticism. We think the novelty of the study lies in three aspects: (1) duration of the exposure, (2) sucralose dosage, and (3) association between sucralose-induced dysbiosis and alteration of glucose and insulin homeostasis. Most of the clinical trials conducted until now have examined the short-term effect of sucralose on the gut microbiota composition using sucralose quantities near the Acceptable Daily Intake (ADI). For instance, in the study mentioned above, Ahmad and coworkers decided to give sucralose in two 2-week periods, incorporating a 4-week washout period and using 13% and 40% of the ADI for sucralose, according to the FAO/JECFA and the FDA, respectively. We opted for a long-term sucralose exposure for 10 weeks, using 48 mg sucralose per day, a much smaller quantity of sucralose than that used by Ahmad et al. that represents less than 5% and 15% of the ADI, according to the FAO/JECFA and the FDA, respectively. Interestingly, Prof. Dr. Xiaofa Qin has recently proposed that the effect of sucralose consumption on human health may exhibit a bell-shaped dose response (J Obes Metab Syndr. 2020 Sep 30;29(3):237-239). Thus, the microbiome composition may follow the same behavior than insulin sensitivity, wherein the most potent effects of sucralose may present at much lower doses than those recommended in the ADI. We think our work enriches the discussion regarding the safety of consuming sucralose quantities near the ADI for short time periods instead of drinking sucralose amounts much less than the ADI for long time periods, which somehow may be closer to the sucralose amount daily consumed the general population. Finally, we compared the long-term effects of sucralose not only on glucose and insulin levels but also on the relative abundance of Bifidobacteriales, Bacteroidales, Lactobacillales, and Eubacteriales, finding that increases in the area under the curve of glucose and insulin maximum peak are linked to changes in the proportion of Lactobacillales and Eubacteriales. We respectfully think that all these aspects strengthen the novelty of the study and following the Reviewer’s observation, we added several discussion paragraphs citing the paper mentioned above and many others, which has enriched the manuscript’s discussion. Please find these changes marked with yellow color at pages 13 and 14.
We sincerely thanks to the Reviewer for her/his very accurate comments and observations that have indubitably improved the last version of this work. Thank you.

Reviewer 3 Report
The topic of effects of sweeteners including sucralose on the intestinal microbiome is the topic of a number of published studies in both human and animal models. One need only do a Google or PubMed search and there are many dozens of studies, many very similar to the present studies. Notably many studies very similar to the studies of the present manuscript was not even cited . Another recent study in mice focused on the liver effects of sucralose inclusion in the mouse diet and relation to the intestinal microbiome, known to be centrally involved in many types of liver inflammation. This citation is https://www.ncbi.nlm.nih.gov/pmc/articles/PMC5522834/. An important and comprehensive review from 2019 was entitled Effect of Sweeteners on the Gut Microbiota: A Review of Experimental Studies and Clinical Trials and this was also not cited in the present manuscript https://www.ncbi.nlm.nih.gov/pmc/articles/PMC6363527/.
The point of the preceding discussion was to let the authors know that their study presently has limited novelty and therefore impact. The studies are done reasonably well, although the methodological limitation that they only analyzed for certain bacterial types limits the ability to use the information and to discover what microbiome changes are going on. With modest additional effort and up to date free computer software, a deeper understanding of the bacterial microbiome changes could have been obtained. The granularity of the microbiome analysis is one of the strengths of the study, but this is subject to interpretation. The authors have begun to determine whether changes in humans response to sucralose in the intestinal biome potentially translated to metabolic changes is dependent on specific biome species. This is interesting and important, but limited data is available to decipher mechanisms. One way to increase the impact is to have more investigation of the roles of Clostridium coccoides or certain Lactobacilli. For the Lactobacilli the authors should confirm which of the revised genus are covered by their primer. For the information on new designation of Lactobacillus by ISAPP and the reference please see https://4cau4jsaler1zglkq3wnmje1-wpengine.netdna-ssl.com/wp-content/uploads/2020/04/Lactobacillus_scientist_linked.pdf.
One manner to increase the impact would be to isolate the Clostridium potentially responsible and provide it to the subjects. One manner to increase the impact of the present studies is to perform a metabolomic analysis of the subjects. This approach may not definitively find the Clostridial or Lactobacillus metabolite that plays a role, but is the first step. To determine causality, the metabolite must be modulated and this could most readily be performed in an animal model, but effects of bacteria on humans are many times different from animals.
For all the attention that the intestinal microbiome has received, instances where physiological effects have been identified to occur due to changes in the bacteria and their structures or metabolites are modest. The same applies to natural dietary products and their potential positive or negative health benefits for the host. Without a physiological effect and linked to a bacteria, mycotic species, or virus of the intestine, and effects of that biome member to a physiological response, the impact is limited and handwaving and hypothesis without experimental approach can only generate modest interest. Why should one be concerned that sucralose might change the intestinal microbiome? This had been studied in a number of papers, but the authors have not referenced much literature on the physiological effects of sucralose in the diet. Metabolic and immune effects of sucralose have been studied most extensively, but there are others, a Google search will reveal these. Thus in addition to not presenting publications with very similar experimentation, the authors have not presented the background why their studied might be important nor have they presented sufficient background on why changes in the bacterial species very modestly effected in the studies, might be important. This text revision would not in the opinion of this reviewer make the manuscript acceptable, more data and experimentation is needed.
MINOR
A minor consideration is the sequence of the pan bacterial primer should be presented.
Author Response
Reviewer #3
Query (Q) 1
The topic of effects of sweeteners including sucralose on the intestinal microbiome is the topic of a number of published studies in both human and animal models. One need only do a Google or PubMed search and there are many dozens of studies, many very similar to the present studies. Notably many studies very similar to the studies of the present manuscript was not even cited. Another recent study in mice focused on the liver effects of sucralose inclusion in the mouse diet and relation to the intestinal microbiome, known to be centrally involved in many types of liver inflammation. This citation is https://www.ncbi.nlm.nih.gov/pmc/articles/PMC5522834/. An important and comprehensive review from 2019 was entitled Effect of Sweeteners on the Gut Microbiota: A Review of Experimental Studies and Clinical Trials and this was also not cited in the present manuscript https://www.ncbi.nlm.nih.gov/pmc/articles/PMC6363527/.
Reply (R) 1
We totally concur with the Reviewer. In fact, following the accurate Reviewer´s observation, we added several paragraphs at the Introduction and Discussion sections, wherein we thoroughly discuss previous evidence regarding the effect of non-caloric artificial sweeteners on the gut microbiome. Please find this information marked with yellow color at pages 2, 13, and 14.
Q2
The point of the preceding discussion was to let the authors know that their study presently has limited novelty and therefore impact. The studies are done reasonably well, although the methodological limitation that they only analyzed for certain bacterial types limits the ability to use the information and to discover what microbiome changes are going on. With modest additional effort and up to date free computer software, a deeper understanding of the bacterial microbiome changes could have been obtained. The granularity of the microbiome analysis is one of the strengths of the study, but this is subject to interpretation. The authors have begun to determine whether changes in humans response to sucralose in the intestinal biome potentially translated to metabolic changes is dependent on specific biome species. This is interesting and important, but limited data is available to decipher mechanisms. One way to increase the impact is to have more investigation of the roles of Clostridium coccoides or certain Lactobacilli. For the Lactobacilli the authors should confirm which of the revised genus are covered by their primer. For the information on new designation of Lactobacillus by ISAPP and the reference please see https://4cau4jsaler1zglkq3wnmje1-wpengine.netdna-ssl.com/wp-content/uploads/2020/04/Lactobacillus_scientist_linked.pdf.
R2
We thank to the Reviewer for her/his criticism. We think the novelty of the study lies in three aspects: (1) duration of the exposure, (2) sucralose dosage, and (3) association between sucralose-induced dysbiosis and alteration of glucose and insulin homeostasis. Most of the clinical trials conducted until now have examined the short-term effect of sucralose on the gut microbiota composition using sucralose quantities near the Acceptable Daily Intake (ADI). For instance, in the study mentioned above, Ahmad and coworkers decided to give sucralose in two 2-week periods, incorporating a 4-week washout period and using 13% and 40% of the ADI for sucralose, according to the FAO/JECFA and the FDA, respectively. We opted for a long-term sucralose exposure for 10 weeks, using 48 mg sucralose per day, a much smaller quantity of sucralose than that used by Ahmad et al. that represents less than 5% and 15% of the ADI, according to the FAO/JECFA and the FDA, respectively. Interestingly, Prof. Dr. Xiaofa Qin has recently proposed that the effect of sucralose consumption on human health may exhibit a bell-shaped dose response (J Obes Metab Syndr. 2020 Sep 30;29(3):237-239). Thus, the microbiome composition may follow the same behavior than insulin sensitivity, wherein the most potent effects of sucralose may present at much lower doses than those recommended in the ADI. We think our work enriches the discussion regarding the safety of consuming sucralose quantities near the ADI for short time periods instead of drinking sucralose amounts much less than the ADI for long time periods, which somehow may be closer to the sucralose amount daily consumed the general population. Finally, we compared the long-term effects of sucralose not only on glucose and insulin levels but also on the relative abundance of Bifidobacteriales, Bacteroidales, Lactobacillales, and Eubacteriales, finding that increases in the area under the curve of glucose and insulin maximum peak are linked to changes in the proportion of Lactobacillales and Eubacteriales. We respectfully think that all these aspects strengthen the novelty of the study and following the Reviewer’s observation, we added several discussion paragraphs citing the paper mentioned above and many others, which has enriched the manuscript’s discussion. Please find these changes marked with yellow color at pages 13 and 14.
Additionally, we sincerely apologize for the misunderstanding in the identification of Lactobacillales. The main goal of the study was to examine the effect of sucralose consumption on the relative abundance of bacterial groups belonging to the phyla Actinobacteria, Bacteroidetes, and Firmicutes, which have been previously identified as potential contributors to metabolic dysfunction. We did not want to explore the whole intestinal microbiome or specific bacterial species and for this reason, we apologize for this misunderstanding. Following the Reviewer´s suggestion and to avoid data misinterpretation, we reformatted the manuscript by analyzing changes in the relative abundance of the order Lactobacillales that belongs to the phylum Firmicutes and includes the genus Lactobacillus (Firmicutes-Lactobacillales), among others. In this way, results are now clearer and more informative. Please see these changes marked with fuchsia color at pages 1, 2, 4, 9-15, and in new Figures 4 and 5, and their figure legends.
Q3
One manner to increase the impact would be to isolate the Clostridium potentially responsible and provide it to the subjects. One manner to increase the impact of the present studies is to perform a metabolomic analysis of the subjects. This approach may not definitively find the Clostridial or Lactobacillus metabolite that plays a role, but is the first step. To determine causality, the metabolite must be modulated and this could most readily be performed in an animal model, but effects of bacteria on humans are many times different from animals.
R3
We thank to the Reviewer for her/his very interesting research proposals that we will take into account in order to increase the impact of this study.
Q4
For all the attention that the intestinal microbiome has received, instances where physiological effects have been identified to occur due to changes in the bacteria and their structures or metabolites are modest. The same applies to natural dietary products and their potential positive or negative health benefits for the host. Without a physiological effect and linked to a bacteria, mycotic species, or virus of the intestine, and effects of that biome member to a physiological response, the impact is limited and handwaving and hypothesis without experimental approach can only generate modest interest. Why should one be concerned that sucralose might change the intestinal microbiome? This had been studied in a number of papers, but the authors have not referenced much literature on the physiological effects of sucralose in the diet. Metabolic and immune effects of sucralose have been studied most extensively, but there are others, a Google search will reveal these. Thus in addition to not presenting publications with very similar experimentation, the authors have not presented the background why their studied might be important nor have they presented sufficient background on why changes in the bacterial species very modestly effected in the studies, might be important. This text revision would not in the opinion of this reviewer make the manuscript acceptable, more data and experimentation is needed.
R4
As mentioned above, we believe all your comments and suggestions are of great relevance and for this reason, we have discussed our own findings in light of previous publications with very similar experimentation. Moreover, we have reformatted the background to clear up the novelty of the study and the importance of examining changes in gut microbiome in response to long-term sucralose consumption. Last but not least, we presented the long-term effects of sucralose consumption not only on glucose and insulin levels but also on the relative abundance of Bifidobacteriales, Bacteroidales, Lactobacillales, and Eubacteriales, finding that increases in the area under the curve of glucose and insulin maximum peak are significantly linked to changes in the proportion of Lactobacillales and Eubacteriales, which as far as we know, is one of the first clinical trials pointing out to a relationship between metabolic dysfunction and changes in gut microbiota. Please see these changes marked with yellow color at pages 2, 10, 12-14, with special emphasis on new Figures 4 and 5, and their figure legends.
We sincerely want to thank you for your research proposals and insights, which will improve the quality of research we try to conduct.
Q5
MINOR
A minor consideration is the sequence of the pan bacterial primer should be presented.
R5
The sequence of the Pan-bacteria universal primer is presented in the Material and Methods section. Please find this information marked with yellow color at page 4, lines 180-181.
We sincerely thank all your comments and observations that have indubitably improved the last version of this work. Thank you.

Round 2
Reviewer 1 Report
Thank you to the authors for considering my previous comments, I appreciated that. Concerning the present manuscript, in my opinion, the authors have an improved manuscript and a better view of their results, I encouraged the authors to take these analyzed data and move forward to MiSeq analyses or similar. The price now is competitive and in some cases rounds around 50-60 euros per sample.
Author Response
Reviewer #1
Query
Thank you to the authors for considering my previous comments, I appreciated that. Concerning the present manuscript, in my opinion, the authors have an improved manuscript and a better view of their results, I encouraged the authors to take these analyzed data and move forward to MiSeq analyses or similar. The price now is competitive and in some cases rounds around 50-60 euros per sample.
Reply
We thank to the Reviewer for her/his kind comments on this new version of the manuscript. To move forward, we will definitely use MiSeq to fully characterize bacterial species that change in response to sucralose ingestion. Of course, this subsequent work will take at least sixteen months to be completed and data resulting from this research branch will be analyzed and communicated after that.
For clarification, we added to the discussion section a sentence pointing out this new research perspective. Please find this information marked with red color at page 15.

Reviewer 3 Report
The authors have made many improvements. Notably the background is much greater and places this small increment of advance in the context of what is known. The authors have also more clearly explained why this small study is important and provides some new information. Importantly they are now more conservative about interpretation of data, how far the limited bacterial data can be extended. The glucose tolerance and insulin values are strong data and this is stated. Interpretation of what the bacterial changes mean is expanded greatly and as honest manuscripts in this discipline accept and state clearly, without much more investigation and with addition of more microbiological data specific to individual types of bacteria, mechanisms will continue to remain elusive and any suggested hypotheses are correlative presently. The challenge now is to develop the tools and models to make additional non-observation and mechanistic advances.
Author Response
Reviewer #3
Query
The authors have made many improvements. Notably the background is much greater and places this small increment of advance in the context of what is known. The authors have also more clearly explained why this small study is important and provides some new information. Importantly they are now more conservative about interpretation of data, how far the limited bacterial data can be extended. The glucose tolerance and insulin values are strong data and this is stated. Interpretation of what the bacterial changes mean is expanded greatly and as honest manuscripts in this discipline accept and state clearly, without much more investigation and with addition of more microbiological data specific to individual types of bacteria, mechanisms will continue to remain elusive and any suggested hypotheses are correlative presently. The challenge now is to develop the tools and models to make additional non-observation and mechanistic advances.
Reply
We thank to the Reviewer for her/his kind comments on this new version of the manuscript. We are now working on a research project that involves humanized mice and seven-day rifaximin treated-subjects, in whom gut microbiota will be reconstituted using different proportions of Lactobacillales and Eubacteriales with the aim of studying the possible mechanisms contributing to altered homeostasis of glucose and insulin in response to sucralose. Of course, this subsequent work will take at least two years to be completed and data resulting from this research branch will be analyzed and communicated after that.
For clarification, we added to the discussion section a sentence pointing out this new research perspective. Please find this information marked with yellow color at page 15.
